# Imaging-based chemical screening reveals activity-dependent neural differentiation of pluripotent stem cells

Yaping Sun[1,2,3†], Zhiqiang Dong[1,2,3†], Taihao Jin[4,5,6], Kean-Hooi Ang[7,8], Miller Huang[9,10,11], Kelly M Haston[4,9], Jisong Peng[1,2,3], Tao P Zhong[12], Steven Finkbeiner[4,5,7,9,13,14], William A Weiss[9,10,11], Michelle R Arkin[7,8], Lily Y Jan[4,5,6], Su Guo[1,2,3,12]*

[1]Department of Bioengineering and Therapeutic Science, University of California, San Francisco, San Francisco, United States; [2]Eli and Edythe Broad Center of Regeneration Medicine and Stem Cell Research, University of California, San Francisco, San Francisco, United States; [3]Programs in Human Genetics and Biological Sciences, University of California, San Francisco, San Francisco, United States; [4]Department of Physiology, University of California, San Francisco, San Francisco, United States; [5]Department of Biochemistry, University of California, San Francisco, San Francisco, United States; [6]Howard Hughes Medical Institute, University of California, San Francisco, San Francisco, United States; [7]Department of Pharmaceutical Chemistry, University of California, San Francisco, San Francisco, United States; [8]Small Molecule Discovery Center, University of California, San Francisco, San Francisco, United States; [9]Department of Neurology, University of California, San Francisco, San Francisco, United States; [10]Department of Neurological Surgery, University of California, San Francisco, San Francisco, United States; [11]Department of Pediatrics, University of California, San Francisco, San Francisco, United States; [12]State Key Laboratory of Genetic Engineering, Department of Genetics, Fudan University School of Life Sciences, Shanghai, China; [13]Keck Foundation Program in Brain Cell Engineering, Roddenberry Center for Stem Cell Biology and Medicine, Gladstone Institutes of Neurological Disease, San Francisco, United States; [14]Taube–Koret Center for Neurodegenerative Disease Research, San Francisco, United States

*For correspondence: su.guo@ucsf.edu

†These authors contributed equally to this work

Competing interests: The authors declare that no competing interests exist.

**Abstract** Mammalian pluripotent stem cells (PSCs) represent an important venue for understanding basic principles regulating tissue-specific differentiation and discovering new tools that may facilitate clinical applications. Mechanisms that direct neural differentiation of PSCs involve growth factor signaling and transcription regulation. However, it is unknown whether and how electrical activity influences this process. Here we report a high throughput imaging-based screen, which uncovers that selamectin, an anti-helminthic therapeutic compound with reported activity on invertebrate glutamate-gated chloride channels, promotes neural differentiation of PSCs. We show that selamectin's pro-neurogenic activity is mediated by γ2-containing GABA$_A$ receptors in subsets of neural rosette progenitors, accompanied by increased proneural and lineage-specific transcription factor expression and cell cycle exit. In vivo, selamectin promotes neurogenesis in developing zebrafish. Our results establish a chemical screening platform that reveals activity-dependent neural differentiation from PSCs. Compounds identified in this and future screening might prove therapeutically beneficial for treating neurodevelopmental or neurodegenerative disorders.

**eLife digest** Pluripotent stem cells have the potential to become most of the cell types that make up an organism. However, the signals that trigger these cells to turn into neurons rather than lung cells or muscle cells, for example, are not fully understood. Proteins called growth factors are known to have a role in this process, as are transcription factors, but it is not clear if other factors are also involved.

In an attempt to identify additional mechanisms that could contribute to the formation of neurons, Sun et al. screened more than 2,000 small molecules for their ability to transform mouse pluripotent stem cells into neurons in cell culture. Surprisingly, they found that a compound called selamectin, which is used to treat parasitic flatworm infections, also triggered stem cells to turn into neurons.

Selamectin works by blocking a particular type of ion channel in flatworms, but this ion channel is not found in vertebrates, which means that selamectin must be promoting the formation of neurons in mice via a different mechanism. Given that a drug related to selamectin is known to act on a subtype of receptors for the neurotransmitter GABA, Sun et al. wondered whether these receptors—known as $GABA_A$ receptors—might also underlie the effects of selamectin. Consistent with this idea, drugs that increased $GABA_A$ activity stimulated the formation of neurons, whereas drugs that reduced $GABA_A$ function blocked the effects of selamectin.

In addition, Sun et al. showed that selamectin triggers human embryonic stem cells to become neurons, and that it also promotes the formation of new neurons in developing zebrafish in vivo. As well as revealing an additional mechanism for the formation of neurons from stem cells, the screening technique introduced by Sun et al. could help to identify further pro-neuronal molecules, which could aid the treatment of neurodevelopmental and neurodegenerative disorders.

## Introduction

Mouse embryonic stem cells (mESCs), capable of generating most cell types that constitute the entire organism, have made important contributions to our understanding of mammalian biology (*Smith, 2001*). How mESCs differentiate into neural lineages is a fascinating question that remains incompletely understood (*Okano and Temple, 2009*; *Gaspard and Vanderhaeghen, 2010*). Neural induction, the first step in neural differentiation of mESCs, requires active FGF signaling (*Ying et al., 2003*) and inhibition of the BMP/TGF-beta pathway (*Chambers et al., 2009*). Subsequent regional identity and lineage-guided differentiation are further regulated by the presence or absence of various morphogens or transcription factors (*Lee et al., 2000*; *Wichterle et al., 2002*; *Andersson et al., 2006*; *Martinat et al., 2006*). However, it is unknown whether mechanisms additional to growth factors and transcription regulators direct the differentiation of mESCs into neural lineages.

Effective means for perturbing a complex biological system are key to gaining new insights into the underlying molecular and cellular mechanisms. Small organic molecules have proven to be invaluable tools for probing biological mechanisms, owing to their versatile nature and ease of application and removal from the system under study (*Stockwell, 2004*; *Zon and Peterson, 2005*). These features also make bioactive small molecules highly attractive for therapeutic applications. One critical challenge in small molecule discovery is that the chemical space is infinite, thereby requiring high throughput screening for speed and bioassays that are of sufficient specificity and sensitivity to distinguish active small molecules from background noise.

Here we report an imaging-based screen of ~2000 bioactive compounds in mESC monolayer cultures labeled with the anti-tyrosine hydroxylase (TH) antibody (a marker for dopaminergic, noradrenergic, and adrenergic neurons). We identified small molecules that increased the appearance of $TH^+$ neurons in the assay, including those with known neurotrophic activity and those that are functionally novel. Notably, we show that the anti-parasitic compound selamectin, with reported activity on invertebrate glutamate-gated chloride channels, increased not only $TH^+$ neurons, but also multiple other neural types including the serotonergic (5-HT), GABAergic, and Islet+ motor neurons as well as Olig2+ oligodendrocytes. We further reveal, through pharmacology, genetics, single-cell electrophysiological recordings and clonal analyses, that selamectin acts by enhancing GABAA receptor signaling, increasing the expression of proneural and lineage-specific transcription factors, and promoting cell cycle exit

and differentiation of neural progenitors. We also demonstrate that selamectin can increase neuronal differentiation in human ESCs and induced pluripotent stem cells (iPSCs) as well as in vivo in the developing zebrafish.

## Results

### High throughput chemical screen

In order to apply chemistry to probe the basic biology of neural differentiation from pluripotent stem cells (PSCs), we designed a high content screen to isolate small molecules that can increase the total number of TH+ neurons derived from mESC monolayer cultures. The system was chosen for several reasons: First, the mESC culture system is an established model for understanding neural development, with much insight gained in recent years (*Okano and Temple, 2009*; *Gaspard and Vanderhaeghen, 2010*). Second, mESCs can be cultured in large quantities and in multi-well plates in a high throughput manner. Finally, our adaptation of the mESC monolayer culture and differentiation method (*Ying and Smith, 2003*) showed that a relatively low and consistent number of TH+ neurons were detected in the culture system (*Figure 1*).

A three-stage protocol was devised ('Materials and methods' for details) (*Figure 1A*). The mESCs of both E14 and 46C lines (*Ying and Smith, 2003*) were used, the latter of which expresses GFP reporter under the control of *sox1* promoter. During stage one, undifferentiated mESCs were cultured on a gelatin-coated surface and in the media without LIF, resulting in neural progenitors that express Sox2, Lmx1a, Nestin, and Sox1 (*Figure 1B*). At stage two, neural progenitors were plated into multi-well plates and treated with chemicals for three days. Finally, chemical treatment was withdrawn and cells were cultured for additional three days before immunostaining with anti-TH antibody (Stage three).

This protocol was further subjected to automation at multiple steps, including cell dispensing into 96-well plate using Thermo Matrix Well Plate, compound distribution into wells using Biomek FXP Laboratory Automation Workstation, immunostaining using Thermo Matrix PlateMate Plus, image capture using GE INCell 1000/2000, and image quantification using INCell Developer software ('Materials and methods' for details). We then screened a library containing 2080 biologically active and structurally diverse compounds, including many FDA approved and currently marketed drugs. Compounds were screened at a final concentration of 1 µM in a volume of 120 µl per well containing 0.67% DMSO (vol/vol). After automated immunostaining, image acquisition, and image analysis, the percentage of TH+ cells in each well was calculated (*Figure 1C*). We did not use actual cell count (as cells in the well are not well separated, making 'cell count' inaccurate); instead, we calculated the area of each segmented target. The percentage of TH signal in each well was expressed as a ratio of TH-covered area over DNA-covered area. The final readout was calculated as fold change compared to the DMSO-treated control. The cut-off for selecting primary hits was set as fold change > mean + 3 S.D. relative to DMSO control, which is a rather stringent selection criteria based on previous studies (*Borowiak et al., 2009*).

To assess assay performance, the coefficient of variation (C.V.) of DMSO control was calculated for each of the twenty-six 96-well plates screened, and all C.V.s but one were smaller than 20%, suggesting an acceptable variation during this cell-based screen (*Figure 1D*). Out of 2080 chemicals screened, 26 led to a fold change of TH+ cells larger than mean + 3 S.D. (1.16%) (*Figure 1E* for an example), and 20 out of the 26 were neither cytotoxic nor auto-fluorescent (*Figure 1F*).

After two rounds of validation, two compounds were selected as hits, yielding an overall hit rate of 0.09%. One identified molecule is Dihydrodeoxygedunin (DOG), which is a natural product with known neurotrophic activity via activating the TrkB receptor and its downstream signaling cascades (*Jang et al., 2010a*). Both DOG and 7,8-dihydroxyflavone (DHF, another selective TrkB agonist [*Jang et al., 2010b*]) increased TH+ cells in mESC cultures, albeit modestly (*Figure 2*). This data suggest that our screen is capable of identifying compounds with neuronal promoting activity.

### Selamectin increases the differentiation of multiple neural lineages from mESCs

The other hit from our screen is selamectin, whose role in promoting ESC differentiation into TH+ neurons is novel, and was selected for further study. We first determined whether selamectin-induced increase of TH+ neurons is selective for these subtypes by immunocytochemistry with the pan-neuronal marker NeuN. Treatment with selamectin increased the percentage of total neurons, compared to the DMSO-treated

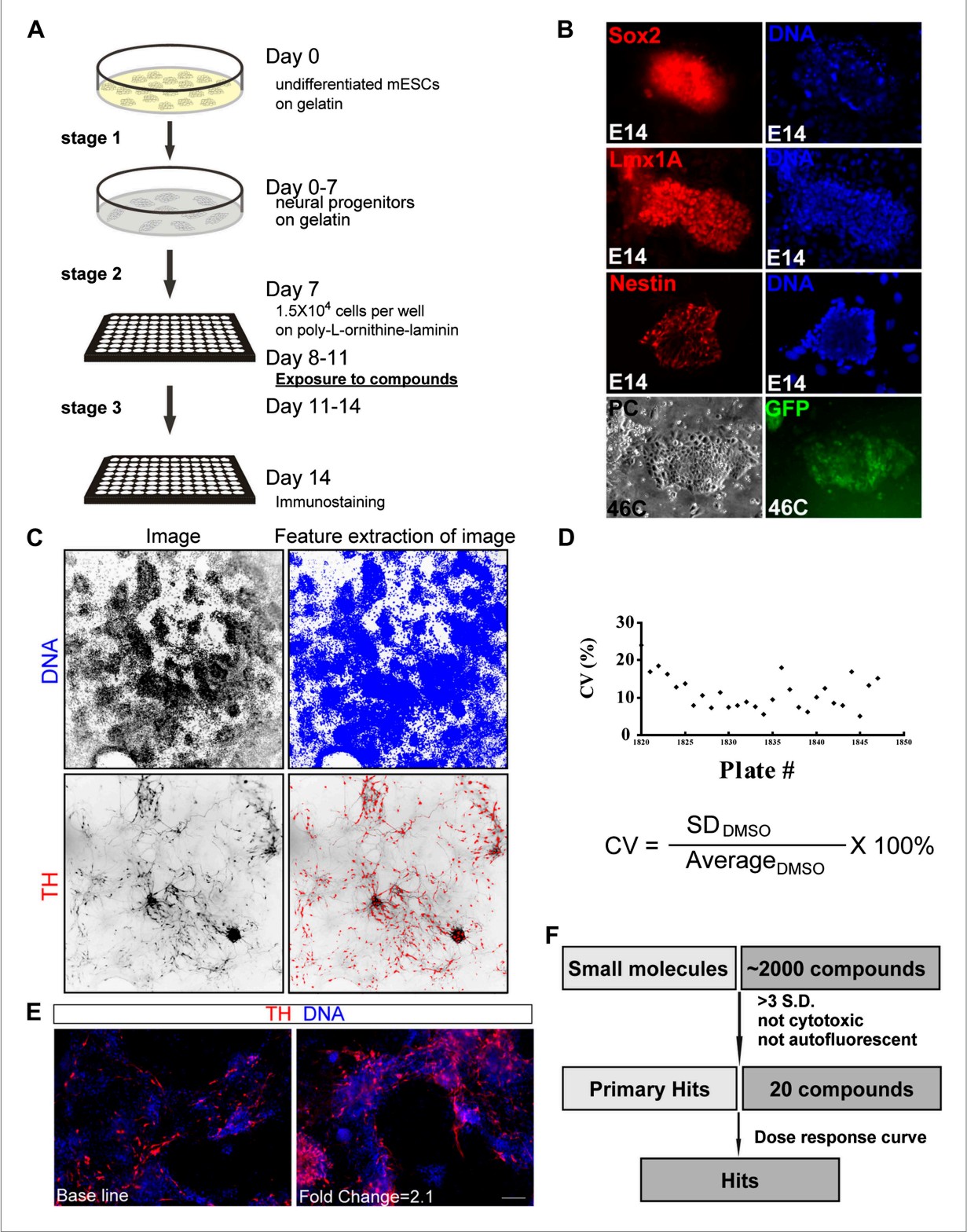

**Figure 1**. High-throughput screening. (**A**) Scheme of the three-stage mESC neuronal differentiation-based chemical screening. (**B**) E14 mESC cultures express neural progenitor makers (Sox2, Lmx1A, and Nestin) after 7-day stage one culture. 46C mESC cultures, in which GFP is driven by the *sox1* promoter, are GFP positive after 7-day stage one culture. (**C**) The quantification analysis of TH⁺ neurons among total cells using the INCell Developer software. (**D**) Summary of coefficient of variation (C.V.) of the DMSO control. (**E**) Representative images of immunostaining in control (left, treated with DMSO) and a hit compound (right). (**F**) A schematic summary of the chemical screen. Scale bar, 10 µm.

$$CV = \frac{SD_{DMSO}}{Average_{DMSO}} \times 100\%$$

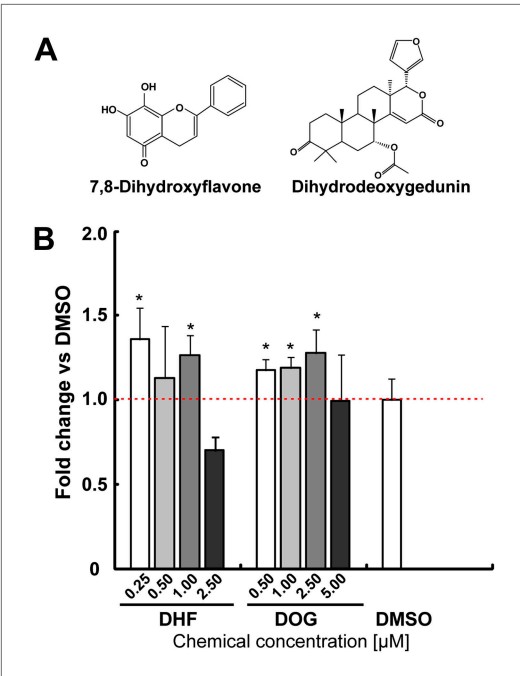

**Figure 2**. The neurotrophin receptor TrkB agonists [Dihydrodeoxygedunin (DOG) and 7,8-dihydroxyflavone (DHF)] increases TH⁺ cells in mESC cultures. (**A**) Structure of DHF and DOG. (**B**) DHF and DOG increase TH% in mESCs (*t* test, p<0.05, n = 4).

control (*Figure 3A–B*). This result suggests that the effect of selamectin is not specific to TH⁺ neuronal subtypes. Further analysis showed that selamectin also significantly increased the production of 5-HT neurons (*Figure 3C*), GABAergic neurons (*Figure 3D*), and Islet⁺ motor neurons (*Figure 3E*). The increase of 5-HT neurons was remarkably high (~sevenfold), suggesting that selamectin might have a preferential activity for inducing 5-HT neurons.

Since neuronal differentiation precedes that of gliogenesis in the vertebrate central nervous system (*Alvarez-Buylla et al., 2001*; *Mehler, 2002*), we wondered whether the increased neuronal production by selamectin is at a cost of later-born glial cells. As our culturing protocol largely favors the differentiation of neurons, we were able to observe only a small numbers of Oligo2⁺ cells at a much later stage (Day 18), suggesting that they were likely to be oligodendrocytes or their precursors. After treatment with selamectin, the percentage of Oligo2⁺ cells also increased compared with the DMSO-treated control (*Figure 3F*), suggesting that the increase of neuronal production is not at the expense of glial production. Taken together, selamectin appears capable of promoting mESC differentiation into both neuronal and oligodendrocyte lineages.

## Dose response and time course of selamectin action

To determine the dose response and time course of selamectin action, we tested a wide range of selamectin concentrations (from 32 nM to 500 nM). The result showed that selamectin functions in a dose-dependent manner with an effector concentration for half-maximum response $EC_{50}$ = 293 nM (*Figure 4A*). When the concentration of selamectin was higher than 500 nM, it became toxic to cells.

Importantly, analysis of the temporal response showed that when cells were treated with selamectin at different stages, its effect was quite different. Treatment during Stage one (from Day 4 to Day 7, mainly composed of undifferentiated ESCs) suppressed cell proliferation, resulting in insufficient number of cells on Day 7 for further evaluation. Treatment during Stage two (from Day 7 to Day 10, mainly composed of neural progenitors) led to a significant increase of neurons evaluated on Day 13 (*Figure 4B*). Shortening the treatment regimen to one day (i.e., from Day 7 to Day 8) or treatment during Stage three (from Day 10 to Day 13, mainly composed of differentiating neurons) showed no significant effect of selamectin (*Figure 4B*). These results suggest that selamectin's proneurogenic activity is likely due to its action on mESC-derived neural progenitors.

## Pharmacological evidence points to an action of selamectin on GABA_A receptors

Selamectin belongs to a chemical family of macrocyclic lactones used to treat nematode infections that cause onchocerciasis (also known as the river blindness) in humans (*Goa et al., 1991*) and as topical or oral parasiticide and antihelminthic on dogs and cats (*Bishop et al., 2000*). This class of drugs disables parasites by displacing glutamate in their muscle synapses through acting on the glutamate-gated chloride (GluCl) channels (*Bloomquist, 2003*), which have no orthologues in vertebrates. A better-studied example of this family is avermectin (for the structures of selamectin and avermectin, *Figure 5A*), which, when tested for its activity in promoting neuronal differentiation, showed a less potent effect than selamectin with a marginal significance achieved at 0.25 μM (*Figure 5B*).

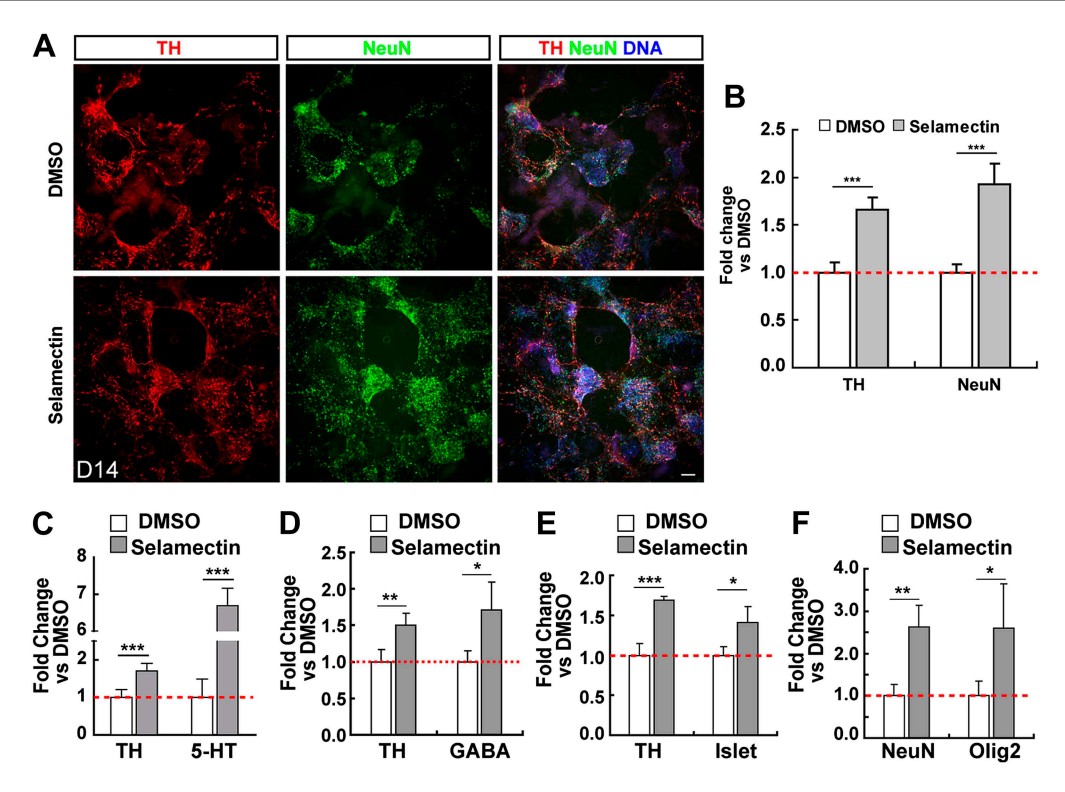

**Figure 3**. Selamectin increases mESC differentiation into multiple neural lineages. (**A**) Representative images of TH and NeuN staining in control (DMSO) and selamectin-treated cultures. (**B**) Quantification shows increased production of both TH+ and total neurons by selamectin (*t*-test, n = 4, p<0.001). (**C–F**) Selamectin treatment also increases the production of 5-HT neurons (**C**), GABAergic neurons (**D**), islet+ motor neurons (**E**), and Olig2+ oligodendocyte precursors (**F**) (*t*-test, n = 4, *p<0.05, **p<0.01, ***p<0.001).

Since GluCl channels are not found in vertebrates, selamectin must act on other protein targets in mESC cultures. Based on gene structure and phylogenetic analyses, vertebrate glycine channels are thought to be orthologous to the invertebrate GluCl channels (***Vassilatis et al., 1997***). It has been reported that ivermectin (an avermectin derivative) can act as an agonist of the glycine receptor (***Shan et al., 2001***). Moreover, non-synaptically released taurine can activate the endogenous glycine receptor during neocortical development (***Flint et al., 1998***). Therefore, we first tested whether taurine, the most abundant endogenous ligand for glycine receptors during neocortical development (***Agrawal et al., 1971***), the deficiency of which affects cortical development in kittens (***Palackal et al., 1986***), has the same proneurogenic effect as selamectin. We treated cells from Day 7 to Day 11 with a wide range of taurine concentrations, none of which, however, showed any effect in increasing TH% among total cells (***Figure 5C***). This result suggests that selamectin does not act through glycine receptors to promote neuronal differentiation from mESCs.

Next we turned to another potential candidate, the GABA$_A$ receptor, since some reports in the literature suggest that avermectin can bind to the GABA$_A$ receptor in rat brain membranes or cultured cerebellar neuronal assays (***Pong et al., 1982***; ***Huang and Casida, 1997***), and avermectin exhibits an anticonvulsant action in a mouse seizure model (***Dawson et al., 2000***). We first tested the effect of the GABA$_A$ receptor agonist Muscimol at a wide range of concentrations (from 1 μM to 100 μM). Remarkably, all of them had proneurogenic activity like selamectin (***Figure 5D***). We also tested the effect of Chlordiazepoxide (CDZ), a positive allosteric modulator of the GABA$_A$ receptor. When cells were treated alone with 10 μM Chlordiazepoxide or 0.3 μM Selamectin, significant proneurogenic activity was observed as compared to the DMSO control (p<0.05, *t*-test, ***Figure 5E***). When cells were treated with both Selamectin (0.3 μM) and CDZ (10 μM), no significant additive effect was observed (p=0.480, *t*-test, ***Figure 5F***), suggesting a lack of synergistic or cooperative action between these two compounds.

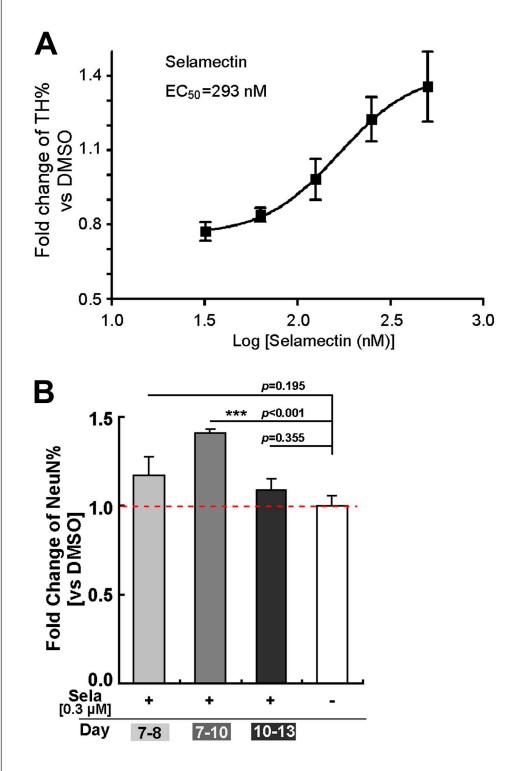

**Figure 4**. Dose response and time course of selamectin's action in mESC cultures. (**A**) Dose response curve of selamectin (based on TH immunostaining). Cells were treated from Day 8 to Day 11. Selamectin is toxic above 500 nM. The EC50 value and curve fitting were performed with GraphPad Prism using a Sigmoidal dose-response (variable slope). Data are presented as mean ± S.D., n = 4. (**B**) Time course of selamectin's action. Only cells treated with selamectin from Day 7 to Day 10 show significant increase of TH+ neurons (*t*-test, n = 4).

To determine whether the proneurogenic activity of selamectin is indeed mediated through the GABA$_A$ receptor, we asked whether it could be blocked by bicuculline (Bicu, GABA$_A$ receptor antagonist), picrotoxin (PTX), or pentylenetetrazol (PTZ) (non-competitive blockers of the GABA$_A$ receptor). When cells were treated alone with these chemicals, no effect on baseline differentiation of TH+ neurons was observed, suggesting that at this stage of the mESC culture, GABA might not be released in sufficient quantities to affect neuronal differentiation (**Figure 5F**, white columns, all control were normalized to fold change = 1, displayed as the red dotted line). When cells were treated with selamectin together with these antagonists or blockers (500 μM PTX, 5 mM PTZ, or 100 μM Bicu), the effect of selamectin was blocked, leading to no significant difference between the treated group and control (**Figure 5F**, gray columns). In contrast, the glycine receptor inhibitor strychnine (STY, 100 μM) failed to block the effect of selamectin (**Figure 5F**). Together, these pharmacological data suggest that selamectin promotes multi-lineage neuronal differentiation from mESCs through the activation of GABA$_A$ receptor signaling.

## The γ2 subunit of GABA$_A$ receptor is required for the proneurogenic activity of selamectin

To determine whether GABA$_A$ receptor is required genetically to mediate the effects of selamectin, we used an EsiRNA approach to perturb the activity of genes encoding various GABA$_A$ receptor subunits. The mammalian CNS expresses nineteen GABA$_A$ receptor subunits (α1-6, β1-3, γ1-3, δ, ε, θ, π, ρ1−3), the combinatorial co-assembly of which enables a potentially enormous molecular and functional heterogeneity of GABA$_A$ receptor subtypes (**Farrant and Nusser, 2005**). Through analyzing the expression profiles of all nineteen receptor subunits in mESC cultures, six genes were identified to display high expression level in mESC-derived neural progenitors (data not shown) and were therefore chosen for esiRNA knockdown. Transfection of gene-specific esiRNAs on Day 6/9 and qRT-PCR analysis on Day 8/11 (**Figure 6A**) showed that esiRNAs targeting the α1 (encoded by the *gabara1* gene), β2 (*gabarb2* gene), γ2 (*gabarg2*), and π (*gabarp* gene) subunits were highly effective in reducing the transcript levels of respective genes (**Figure 6B**). Furthermore, treatment with the esiRNA targeting *gabarg2* abolished the effect of selamectin in inducing NeuN+ and TH+ neurons (**Figure 6C–H**). Treatment with the esiRNAs targeting *gabara1*, *gabarb2*, and *gabarp* genes did not noticeably block the effect of selamectin, suggesting that either the reduction of transcript levels is not sufficient to abrogate their gene activity or these subunits do not mediate the effect of selamectin. Together, these results provide genetic evidence that selamectin acts through the γ2-containing GABA$_A$ receptor to promote neuronal differentiation.

## A subset of neural rosette cells in mESC cultures respond to GABA and selamectin

Opposite to its inhibitory roles in adult neurons, GABA signaling mediated by the activation of GABA$_A$ receptors can depolarize cells in the ventricular zone of the rat embryonic cortex (**LoTurco et al.,**

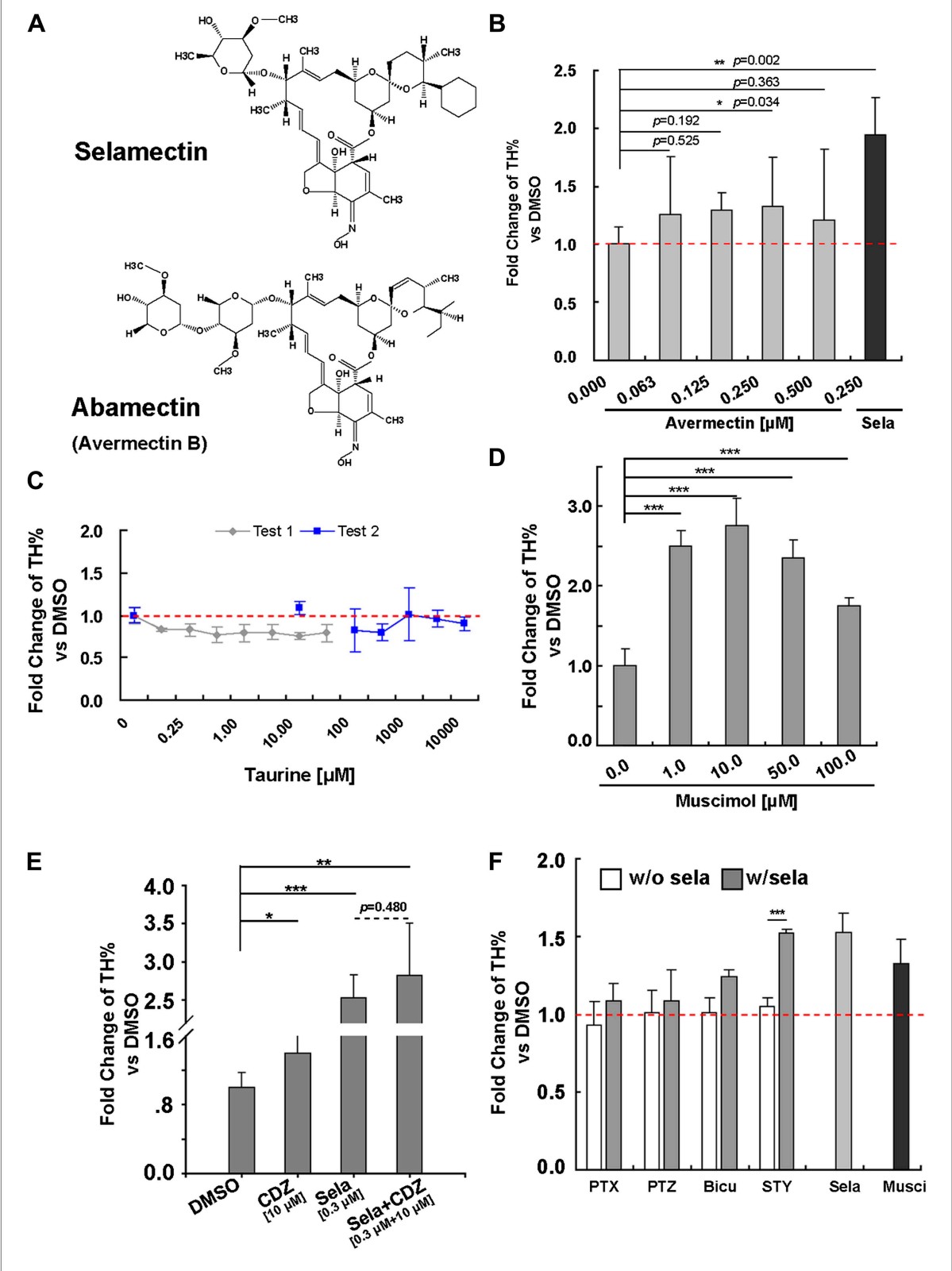

**Figure 5**. Pharmacological evidence indicates that selamectin's proneurogenic activity is mediated by GABA$_A$ receptors. (**A**) Selamectin and avermectin belong to a chemical family of macrocyclic lactones and have the same structural backbone. (**B**) Avermectin has less potent proneurogenic activity than selamectin. Only 250 nM avermectin shows a significant effect, which was much less potent compared to that of 250 nM selamectin (*t*-test, n = 4, p=0.034

*Figure 5. Continued on next page*

*Figure 5. Continued*

for Avermetin vs p=0.002 for selamectin). (**C**) Taurine, the most abundant endogenous ligand for glycine receptors during neocortical development has no proneurogenic activity in mESC cultures. (**D**) Muscimol, a GABA$_A$ receptor agonist, has a significant proneurogenic activity (*t*-test, n = 4, p<0.001). (**E**) Chlordiazepoxide (CDZ), a positive allosteric modulator of GABA$_A$ receptor, also has a significant proneurogenic activity (*t*-test, p<0.05), but there was no obvious additive effect when cells were treated with both selamectin and CDZ (*t*-test, p=0.480). (**F**) The GABA$_A$ receptor antagonist bicucullin and non-competitive blockers picrotoxin and pentylenetetrazol alone had no effect on neuronal production (white columns, control group were normalized to fold change = 1, displayed as the red dot line). However, when tested together with selamectin, the effect of selamectin was blocked (gray columns). In contrast, the glycine receptor inhibitor strychnine does not block the effect of selamectin. Final concentration: Bicuculline (Bicu) = 100 μM; Picrotoxin (PTX) = 500 μM; Pentylenetetrazol (PTZ) = 5 mM; Strychnine (STY) = 100 μM; Selamectin (Sela) = 0.3 μM; Muscimol (Musci) = 10 μM (*t*-test, n = 4).

*1995*), neural progenitor/stem cells in the postnatal mouse sub-ventricular zone (*Liu et al., 2005*) or sub-granular zone (*Tozuka et al., 2005*), due to the elevated internal Cl$^-$ concentrations in progenitor cells and young neurons (*Owens and Kriegstein, 2002*; *Spitzer, 2006*; *Ben-Ari et al., 2007*; *Ge et al., 2007*). Based on this information, we sought to identify the cell types in mESC cultures that might be responsive to GABA and selamectin by performing single-cell electrophysiological recordings of neural activity.

The mESC cell line 46C that expresses GFP driven by *Sox1* promoter was used. Based on the observed temporal effect of selamectin in mESC cultures, we focused our attention on neural rosettes, a functionally distinct type of neural stem cells suggested to represent the earliest NSC stage in vivo (*Elkabetz et al., 2008*). Neural rosettes are easily recognizable because of their characteristic bipolar morphology and radial floral-like arrangement. For the majority of neural rosettes, we also verified their GFP signals (*Figure 7A*). In addition, we examined some non-rosette cells with the morphology of young neurons (*Figure 7E*). For a majority of cells recorded, we tested both GABA and selamectin-induced currents.

Both neural rosettes and non-rosette cells displayed similar passive membrane properties. For neural rosettes, we observed membrane capacitance $C_m$ = 9.4 ± 12.5 pF (mean ± S.D., n = 106), input resistance $R_m$ = 772.5 ± 1037.1 MΩ (mean ± S.D., n = 106), and resting membrane potential $E_{Rev}$ = − 20.9 ± 8.3 mV (mean ± S.D., n = 71). For non-rosette cells, we observed membrane capacitance $C_m$ = 9.5 ± 6.2 pF (mean ± S.D., n = 74), input resistance $R_m$ = 1.2 ± 1.2 GΩ (mean ± S.D., n = 74), and resting membrane potential $E_{Rev}$ = − 22.4 ± 9.2 mV (mean ± S.D., n = 63). The resting membrane potentials of both groups of cells are similar to what has been observed in neural progenitors using similar internal and external recording solutions, either in acute slices (*Wang et al., 2003*) or in vitro (*Stewart et al., 2002*).

To test whether neural progenitors in neural rosettes respond to GABA, we measured the whole cell currents upon GABA applications when membrane potentials of the cells were held at −70 mV. In approximately 21% of the cells (24 out of 111), applications of GABA induced inward currents. The GABA induced currents are sensitive to bicucculin (Bicu, 200 μM), but in some cases the application of 200 μM Bicu did not completely block the GABA induced currents (*Figure 7B*). To determine whether selamectin regulates membrane electrophysiological properties in neural rosette cells, we analyzed responses of the whole-cell currents to the applications of selamectin. Selamectin indeed induced inward currents in neural rosette cells (*Figure 7C*), but the response was heterogeneous. We grouped the cells into four groups based on whether they displayed detectable GABA and selamectin-induced currents. They were GABA$^+$;Sela$^+$ (the cells that displayed both GABA and selamectin induced currents), GABA$^+$;Sela$^-$ (the cells that displayed only GABA induced currents), GABA$^-$;Sela$^+$ (the cells that displayed only selamectin induced currents), and GABA$^-$;Sela$^-$ (the cells that displayed neither GABA nor selamectin induced currents). The numbers of cells belonging to each group were shown in *Figure 7D*. Among the 77 recorded cells, the largest group (43 cells, 64.2%) was those that did not display induced currents by either GABA or selamectin. The second largest group (14 cells, 20.9%) is those that displayed induced currents only by GABA. The third group (10 cells, 14.9%) was those that displayed both GABA and selamectin induced currents. No cells displayed only selamectin-induced currents.

Slightly higher percentage (27 out of 73 cells, 37%) of non-rosette cells (*Figure 7E*) displayed GABA induced currents, which could be blocked by Bicu (*Figure 7F*). Examples of the cells that displayed (upper trace) or did not display (lower trace) selamectin-induced currents were shown in *Figure 7G* and their frequency distribution was shown in *Figure 7H*. Out of 44 non-rosette cells that we tested both GABA and selamectin induced currents, the largest group (20 cells, 45.5%) was the one that

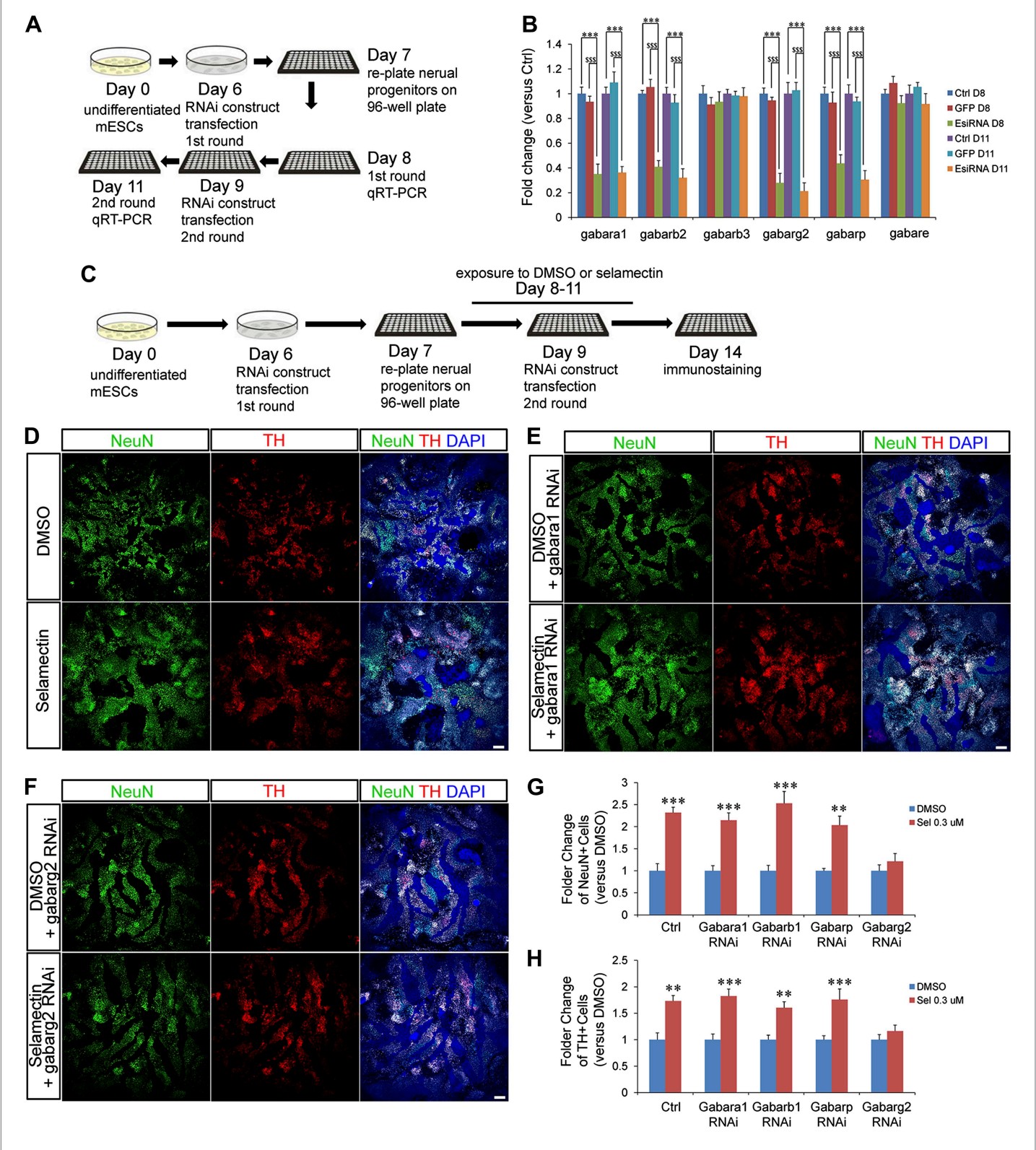

**Figure 6**. Genetic evidence indicates that selamectin's proneurogenic activity is mediated by the γ2 subunit-containing GABA$_A$ receptor. (**A**) Scheme of the EsiRNA transfection and cell harvest for qRT-PCR. (**B**) qRT-PCR shows fold change of the expression of different GABA$_A$ receptor subunits after EsiRNA transfection. ***p<0.001 vs non-transfection control, $$$p<0.001 vs GFP RNA transfection control. (**C**) Scheme of the EsiRNA knockdown experiment to
*Figure 6. Continued on next page*

*Figure 6. Continued*

identify the GABA$_A$ receptor subunit that mediates selamectin's activity. (**D–F**) Representative images of NeuN and TH staining in DMSO and selamectin-treated cultures of non-transfected (**D**), α1 EsiRNA transfected (**E**), and γ2 EsiRNA transfected (**F**). (**G**) Quantification shows knockdown of γ2 subunit but not α1β1 and π subunits abolishes the effect of selamectin in increasing neurons (n = 4, **p<0.01, ***p<0.001 vs DMSO). (**H**) Quantification shows knockdown of γ2 subunit but not α1β1 and π subunits abolishes the effect of selamectin in increasing TH$^+$ neurons neurons (n = 4, **p<0.01, ***p<0.001 vs DMSO). Scale bar, 100 µm.

displayed only GABA-induced currents. The second group (17 cells, 38.6%) was the one that displayed neither GABA nor selamectin induced currents. The third group (7 cells, 15.9%) displayed both GABA- and selamectin-induced currents. Similar to neural rosette cells, no non-rosette cells displayed only selamectin-induced currents. Taken together, our results uncover considerable heterogeneity in the response of neural rosettes and young neurons to GABA or selamectin and indicate that a subset of GABA-responsive neural rosette cells in mESC cultures respond to selamectin.

## Selamectin decreases proliferation through increased expression of proneural and lineage-associated transcription factors

Although GABA-induced depolarization in postnatal neural progenitors causes reduced proliferation, its effect on neurogenesis has been controversial. While one study shows that GABA could promote neuronal differentiation in adult hippocampal progenitor cells (*Tozuka et al., 2005*), another found an inhibitory role on neuronal production in the postnatal sub-ventricular zone (*Liu et al., 2005*). To further probe the mechanisms by which selamectin promotes neuronal differentiation, we performed BrdU incorporation experiment in mESC cultures on Day 11 after the treatment with selamectin for 4 days. Significantly fewer BrdU$^+$ cells were detected in selamectin-treated groups, suggesting decreased proliferation (*Figure 8A–B*).

Since electrical activity in neural progenitor cells can lead to neurotransmitter re-specification through regulating transcription factor expression (*Spitzer, 2012*), we asked whether selamectin exerts an effect on the expression of transcription factors that regulate neuronal specification. At Day 8, cells were sampled for qRT-PCR while the rest were split into two groups for further treatment with either DMSO or selamectin. The gene expression levels were examined by qRT-PCR on Day 11 and the results were compared to that of Day 8. We detected an increased expression of the proneural genes *ascl1* and *neurod*, as well as the DA lineage- associated genes *lmx1a*, *lmx1b*, and *nurr1* in Day 11 culture compared to Day 8 culture, and such increase was further enhanced by selamectin (*Figure 8C*). These results are consistent with the idea that selamectin activates GABA$_A$ receptor and enhances the electrical activity of neural progenitors, thereby increasing the expression of proneural and lineage-associated transcription factors, leading to increased neuronal differentiation.

We also tested whether decreased cell death (or improved neuronal survival) might also contribute to increased neurons detected in selamectin-treated mESC cultures. Cells were treated with selamectin or DMSO from Day 8 to Day 11 and cultured for 4 more days. On Day 14, TUNEL labeling was carried out. The result showed no significant difference in cell death between control and selamectin-treated cultures (*Figure 8D–E*), suggesting that alteration in cell death contributes little to selamectin's pro-neurogenic activity.

## Clonal analysis of mESC-derived neural progenitor cells reveals that selamectin promotes progenitor cell cycle exit toward terminal differentiation

Selamectin-induced decrease of proliferation in mESC cultures could be due to either prolonged cell cycle length or increased cell cycle exit. To further probe the underlying cellular mechanisms, we carried out clonal analysis. Day 6 mESC cultures were transfected with *pCAG-GFP* that resulted in sparse labeling of clonally related cells (*Figure 9A*). Cultured cells were treated with DMSO or 0.3 µM selamectin from Day 8 to Day 11. Time-lapse live imaging was performed on sparsely labeled neural progenitors from Day 8 to Day 14, followed by fixation and immunostaining (*Figure 9A*). We found that the cell cycle length exhibited no significant difference between DMSO- (25.61 ± 6.99 hr, n = 31) and selamectin-treated cells (24 ± 6.53 hr, n = 28) (*Figure 9B*, left panel, *Videos 1, 2*). We also quantified the number of cells within single clones and detected no difference between DMSO- (13.46 ± 4.14 cells per clone, n = 56 clones) and selamectin- (15.15 ± 4.67 per clone, n = 59 clones) treated groups (*Figure 9B*, right panel).

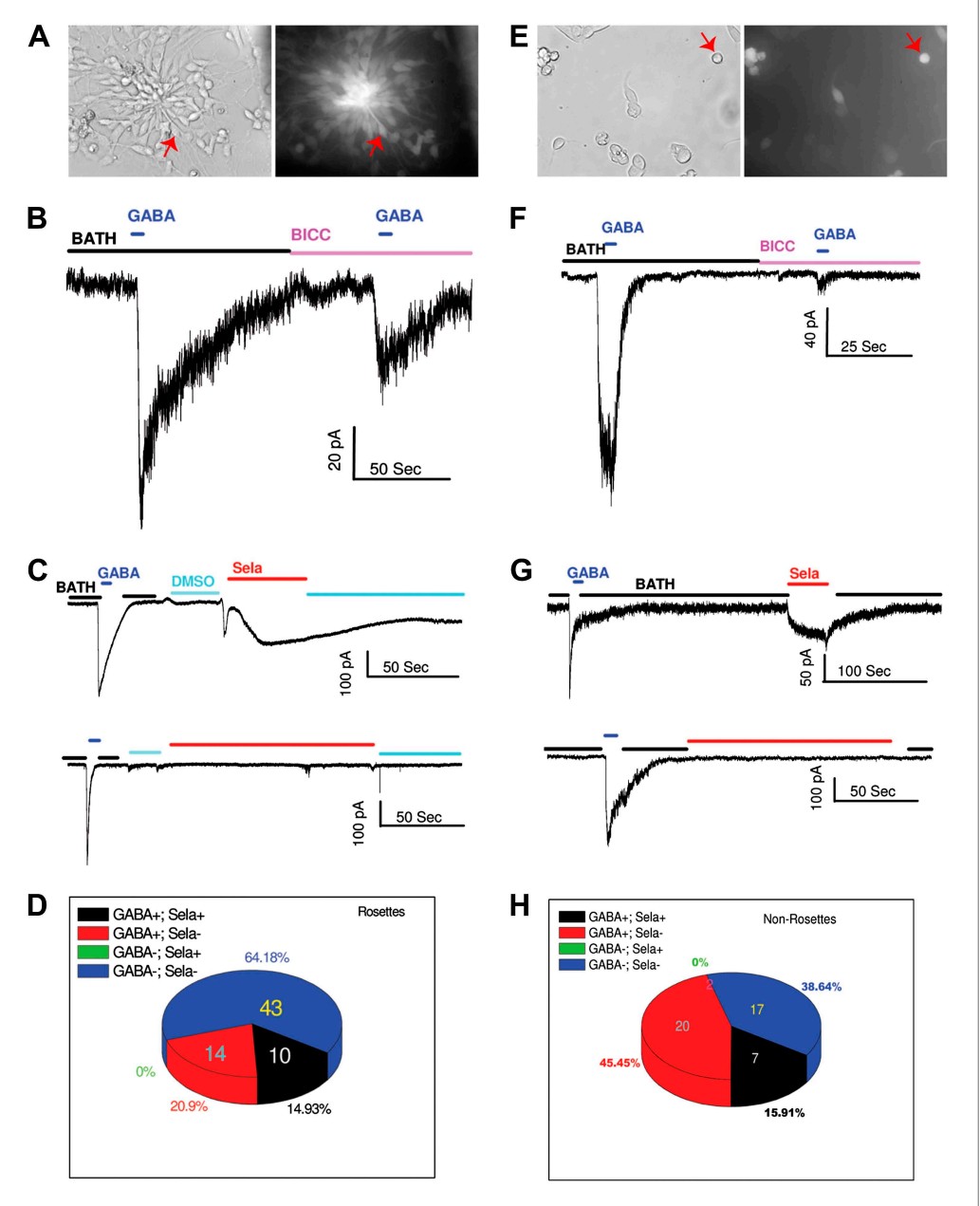

**Figure 7**. GABA and selamectin (Sela)-induced currents in neural rosettes and non-rosette cells. (**A**) Bright field (BF) image (left) and green fluorescent protein (GFP) signal (right) of neural rosette cells. Majority of the cells in the view are neural rosette cells, and one of them with typical morphology is indicated by a red arrow. (**B**) An example of reduction of GABA (500 µM) induced currents by bicuculline (BICC, 200 µM). (**C**) Upper trace, an example of neural rosette cell displaying GABA (100 µM) induced currents that also displayed selamectin induced current. Lower trace, an example of neural rosette cell displaying GABA (100 µM) induced currents that did not display selamectin induced current. 16 µM Selamectin solution contains 0.4% DMSO, therefore 0.4% DMSO containing bath solution was used as the control solution as indicated in the figure. (**D**) A pie chart of the numbers of the four groups of neural rosette cell based on whether they displayed GABA and selamectin induced currents. The four groups are: 1) GABA⁺; Sela⁺, 2) GABA⁺; Sela⁻, 3) GABA⁻; Sela⁺, 4) GABA⁻; Sela⁻. The number of the cells in each group is indicated in the figure, except the group GABA⁻; Sela⁺, which is zero. (**E**) Bright field (BF) image (left) and green fluorescent protein (GFP) signal (right) of non-rosette cells. One cell with typical young neuron morphology is indicated by a red arrow. (**F**) Example of the inhibition of GABA (100 µM) induced currents by bicuculline (BICC, 100 µM). (**G**) Upper trace, an example of non-rosette cell displaying GABA (4 mM) induced current that also displayed
*Figure 7. Continued on next page*

*Figure 7. Continued*

salemectin (8 µM) induced current. Lower trace, an example of non-rosette cell displaying GABA (400 µM) responsive induced current that did not display salemectin (16 µM) induced current. (**H**) A pie chart of the numbers of four groups of non-rosette cells based on whether they displayed GABA and selamectin induced currents. The four groups are: 1) GABA$^+$; Sela$^+$, 2) GABA$^+$; Sela$^-$, 3) GABA$^-$; Sela$^+$, 4) GABA$^-$; Sela$^-$. The number of the cells in each group is indicated in the figure, except the group GABA$^-$; Sela$^+$, which is zero. The application time courses of the control solution (BATH), GABA, bicuculline (BICC) and selamectin (Sela) are indicated by horizontal bars in the figure. The membrane potential was holding at −70 mV in all recordings. The scales for time (horizontal) and currents (vertical) are indicated for each recording in the figure.

The results suggest that selamectin does not change the cell cycle length or the proliferation rate of mESC-derived neural progenitors.

To determine the cell fates within single clones, we performed co-immunostaining of GFP and NeuN (or TH) as well as the proliferation marker Ki67 (*Figure 9C–E*). Clonal quantification of cell fates showed a significant increase of NeuN$^+$ (or TH$^+$) neurons and a concurrent decrease of Ki67$^+$ progenitor cells within single clones treated with selamectin (*Figure 9F*). Taken together, these findings suggest that selamectin acts to promote progenitor cell cycle exit toward terminal differentiation.

## Selamectin promotes the differentiation of multiple neuronal lineages from human pluripotent stem cells (hPSCs)

In order to test the effects of selamectin on neuronal differentiation from hPSCs, we devised a three-stage neuronal differentiation protocol and used the H9 line of hESC (*Thomson et al., 1998*) and a human induced pluripotent stem cell line (hiPSC) (*Kreitzer et al., 2013*) (*Figure 10A*). In agreement with the results from mESCs, treatment with selamectin (0.25 µM, 0.5 µM, 0.75 µM) significantly increased the percentage of total neurons, TH and 5-HT neurons compared to the DMSO-treated control in both H9 hESCs (*Figure 10B–C*) and hiPSCs (*Figure 10D–E*).

## Selamectin promotes neurogenesis in vivo in the developing zebrafish brain

To determine whether selamectin has a pro-neurogenic activity in developing embryos in vivo, we tested its effect on zebrafish. A transgenic line carrying the HuC:GFP transgene (*Park et al., 2000*) was used. HuC/ELAVL3 is a neuron-specific RNA binding protein that is expressed in most differentiating neurons. We treated embryos with selamectin starting at ~75% epiboly (8 hr post fertilization, hpf), when the neuro-ectodermal fate has been determined. The treatment of selamectin lasted until 22 hpf, when nascent neurons began to emerge but not yet became too numerous to quantify (*Figure 11A*). We noted an obvious increase of the overall HuC-GFP$^+$ neurons in the selamectin-treated embryos as compared to the DMSO control (*Figure 11B*). Quantification of the midbrain cluster (boxed) showed a highly significant difference between the selamectin-treated and the control (*Figure 11C*, n =20, *t*-test, p<0.001).

We also examined the effect of selamectin on DA neurons, by treating embryos with selamectin from 8 hpf to 48 hpf followed by immunostaining with the anti-TH antibody (*Figure 11A*). The number of ventral forebrain DA neurons was significantly increased in selamectin-treated embryos as compared to the DMSO control (*Figure 11D–E*). These results suggest that selamectin has pro-neurogenic activity in vivo.

## Discussion

In this study, we have undertaken a chemical genetic approach to identify new mechanisms that regulate neuronal differentiation from PSCs. Three significant advancements are reported here: First, we have established an imaging-based high content small molecule screening method for identifying chemicals that increase the production of TH$^+$ DA neurons from mESCs. We have further shown that compounds identified through mESC-based screening are applicable to hPSCs. Second, It has been previously reported that mild electrical stimulation influences mESCs to assume a neuronal fate (*Yamada et al., 2007*). Through the identification of selamectin, we reveal a novel mechanism underlying the activity-dependent regulation of neuronal differentiation from PSCs. Third, by employing single-cell electro-physiological recordings of mESC-derived neural rosette cells, we uncover for the first time the

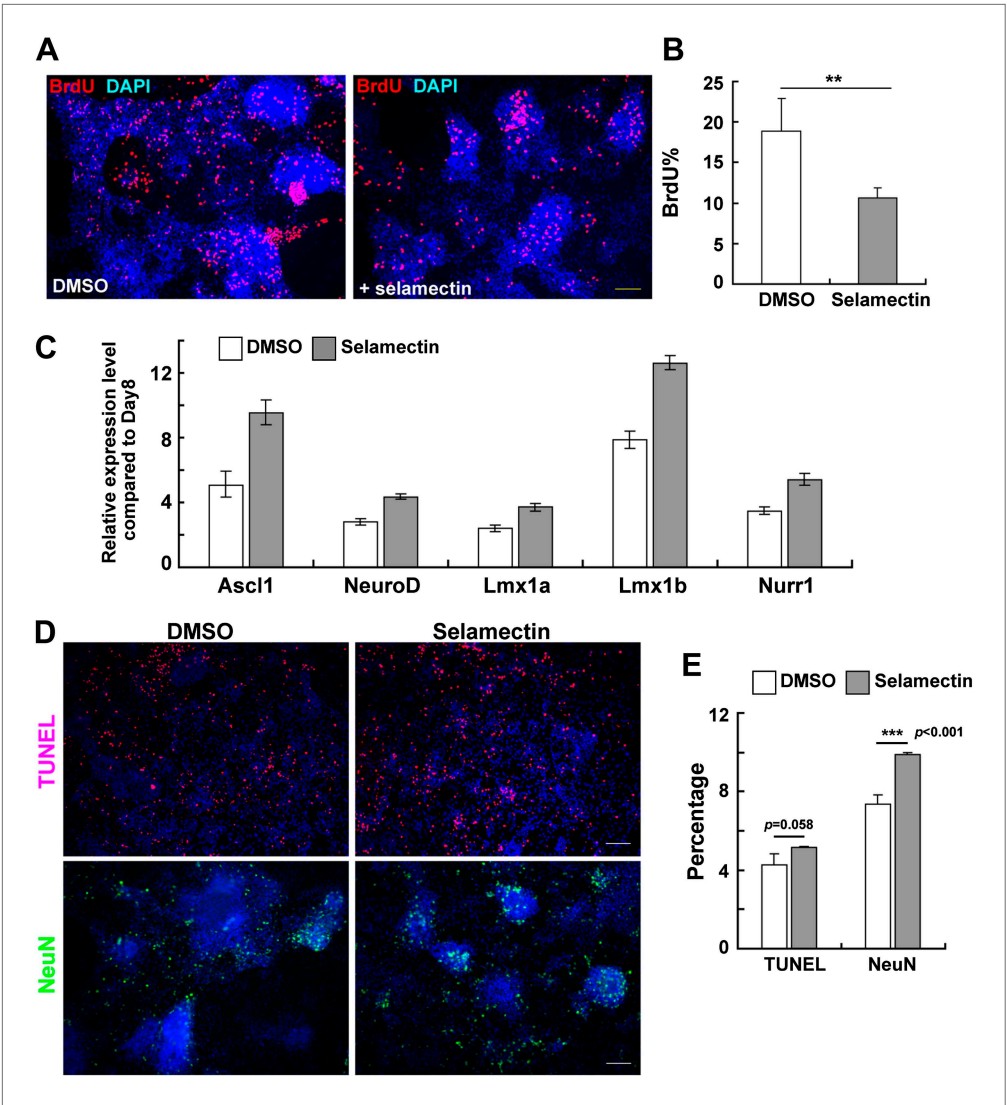

**Figure 8**. Selamectin decreases proliferation and increases the expression of proneural and lineage-associated transcription factors. (**A**) Representative fields show the BrdU incorporation on Day 11 after cells were treated with selamectin (right panel) or DMSO (left panel) for 4 days. Significantly fewer BrdU+ cells were detected in the selamectin-treated group. (**B**) Quantification as the percentage of BrdU+ cells among total cells (*t*-test, n = 4, p=0.008). (**C**) qRT-PCR detects increased expression of proneural (*Ascl1, NeurD*) and lineage-associated transcription factors (*Lmx1a, Lmx1b* and *Nurr1*) in selamectin-treated group. β-actin was used as an input control and data was normalized to expression level on Day 8. (**D**) Representative fields show the TUNEL staining on Day 14 in cell cultures treated with DMSO or selamectin, with NeuN staining as a control to confirm selamectin efficacy in this experiment. (**E**) Quantification shows significant difference in NeuN% (p<0.001) and no significant difference in TUNEL% (p=0.058) (*t*-test, n = 4). Scale bar, 10 μm.

heterogeneity of neural progenitor responses to GABA and selamectin, which has provided a plausible explanation for paradoxical observations of GABA's effects on in vivo neurogenesis (*Liu et al., 2005*; *Tozuka et al., 2005*; *Song et al., 2012*).

Compared to previous high content chemical screening for neuronal enhancers in embryonic stem/progenitor cultures (*Ding et al., 2003*; *Saxe et al., 2007*; *Desbordes et al., 2008*; *Zhou et al., 2010*), the advances of our method are the following: First, Instead of using culture systems that require either feeder cells or neurosphere formation, our assay uses the monolayer differentiation method (*Ying and Smith, 2003*), thus is simple and easy to carry out. Second, the use of monolayer differentiation also renders the background noise fairly low, with baseline TH+ cells of 0.5–3% in control (DMSO) conditions.

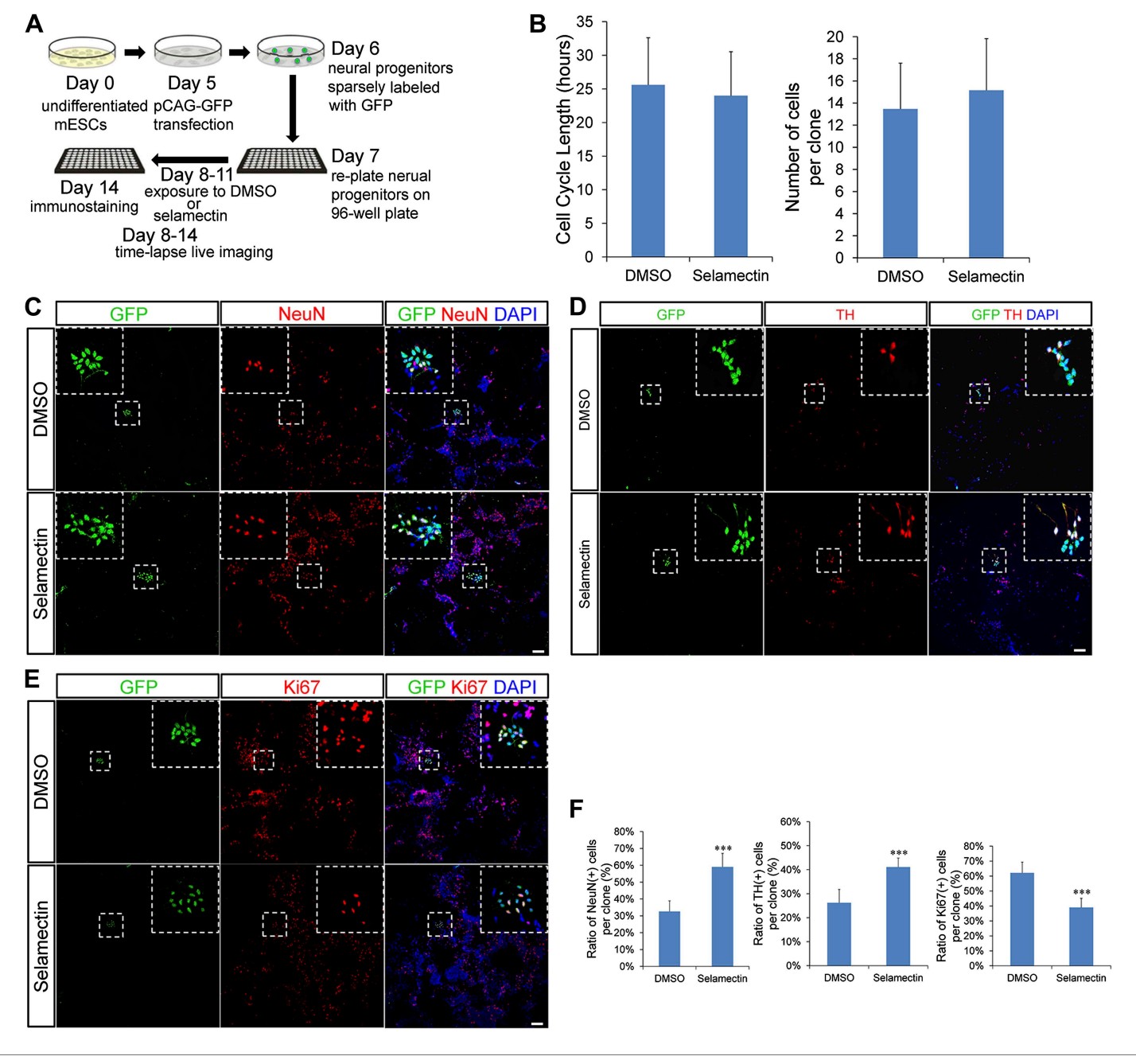

**Figure 9**. Clonal analysis reveals that selamectin promotes progenitor cell cycle exit toward terminal differentiation. (**A**) Scheme of clonal culture analyses. (**B**) Quantification of cell cycle length and number of cells within single clones. (**C–E**) Representative images of double immunostaining of GFP with NeuN (**C**), TH (**D**), or the proliferation marker Ki67 (**E**) within single clones. (**F**) Quantification of the percentage of NeuN+ (left, DMSO n = 49, selamectin n = 51), TH+ (middle, DMSO n = 6, selamectin n = 7), and Ki67+ cells (right, DMSO n = 46, selamectin n = 41) within single clones. ***p<0.001 vs DMSO). Scale bar, 100 μm.

Third, compared to previous screening that uses either pluripotency markers (e.g., Oct4) or general neuronal differentiation markers (e.g., a-tubulin or TuJ1), we used the neuronal subtype specific marker TH. The small number of TH+ cells makes the quantification convenient and accurate, thereby significantly increasing the sensitivity of our assay. Finally, our assay is the first that offers the prospect for identifying compounds that regulate neuronal subtype differentiation, since previous assays use only general pluripotency or neuronal markers.

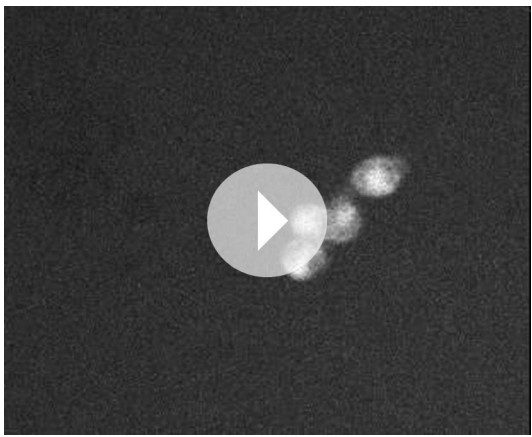

**Video 1**. Time-lapse of a single GFP labeled neural progenitor derived from mESC. The progenitors were treated with DMSO from Day 8 to Day 11. The interval between each frame is 2 hr.

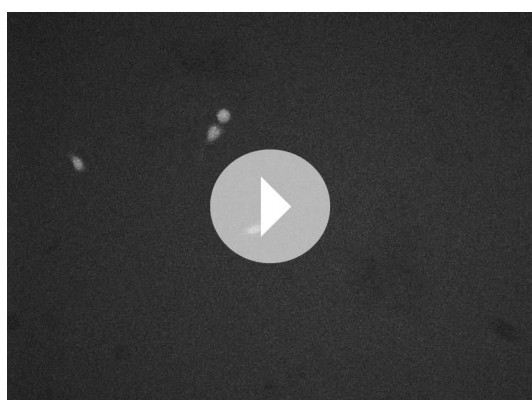

**Video 2**. Time-lapse of a single GFP labeled neural progenitor derived from mESC. The progenitors were treated with 0.3 µM Selamectin from Day 8 to Day 11. The interval between each frame is 2 hr.

This screening platform has enabled us to identify selamectin and show that it increases the production of multiple neuronal lineage types including DA, 5-HT, GABA, and Islet[+] neurons as well as olig2[+] oligodendrocytes. Selamectin is known to target invertebrate glutamate-guided chloride channels that have no orthologues in vertebrates. Our pharmacological, genetic and clonal studies provide first evidence that selamectin targets the γ2 subunit-containing GABA$_A$ receptor to promote progenitor cell cycle exit toward terminal differentiation. Single-cell electrophysiological recordings further show that a subset of neural rosette cells responds to GABA and selamectin via functional GABA$_A$ receptors. The blocking of GABA-induced current by bicuculline (the GABA$_A$ receptor antagonist), however, is incomplete, suggesting that GABA receptors other than GABA$_A$ are also expressed in neural rosette progenitors. Expression of functional GABA$_B$ receptors has been demonstrated in ES cells (*Schwirtlich et al., 2010*). Interestingly, many recorded neural rosette cells display very low membrane input resistance (less than 100 MΩ), and high membrane capacitances (up to 50 pF). Furthermore, applications of 100 µM meclofenamic acid (MFA) (*Liu et al., 2005*) could reversibly reduce the inward currents when the membrane potential of the cells are held at −70 mV. These observations suggest that neural rosette cells may be electrically connected, similar to neural progenitor cells in the ventricular zone of embryonic brains (*LoTurco et al., 1995*) and those in the sub-ventricular zone of adult brains (*Liu et al., 2005*). Thus, it is possible that functional GABA receptors expressed in one cell may render other electrically connected cells to display apparent GABA-induced currents.

Our data reveal that selamectin increases the expression of proneural and lineage-specific transcription factors while reducing proliferation in mESC cultures. Clonal analysis and time-lapse live imaging further uncover the role of selamectin (hence, likely GABA) in promoting progenitor cell cycle exit toward terminal differentiation. These molecular and cellular findings, together with the pharmacogenetic and electrophysiological studies, lead us to propose that selamectin-potentiated activation of the γ2-containing GABA$_A$ receptor in mESC-derived neural progenitor cells causes Cl$^-$ outflow (due to the high internal chloride levels in these progenitor cells), thereby leading to the depolarization of neural progenitors and calcium influx. This further activates proneural and lineage-specific transcription factors, which have established roles in promoting cell cycle exit toward terminal neuronal differentiation (*Bertrand et al., 2002*) (*Figure 8F*). Such neural activity-dependent regulation of transcription factor expression has been reported in developing *Xenopus* embryos (*Demarque and Spitzer, 2010*; *Marek et al., 2010*). Our findings suggest that this is an evolutionarily conserved phenomenon.

GABA signaling influences embryonic cortical neural progenitor proliferation (*Owens and Kriegstein, 2002*) and regulates adult neurogenesis (*Ge et al., 2007*). Paradoxically, either an increase or a decrease of neurogenesis by GABA activation has been observed (*Liu et al., 2005*; *Tozuka et al., 2005*). Recently, in the context of adult neurogenesis, GABA released by local interneurons has been shown to promote the exit of adult neural stem cells from quiescence (thereby promote their proliferation)

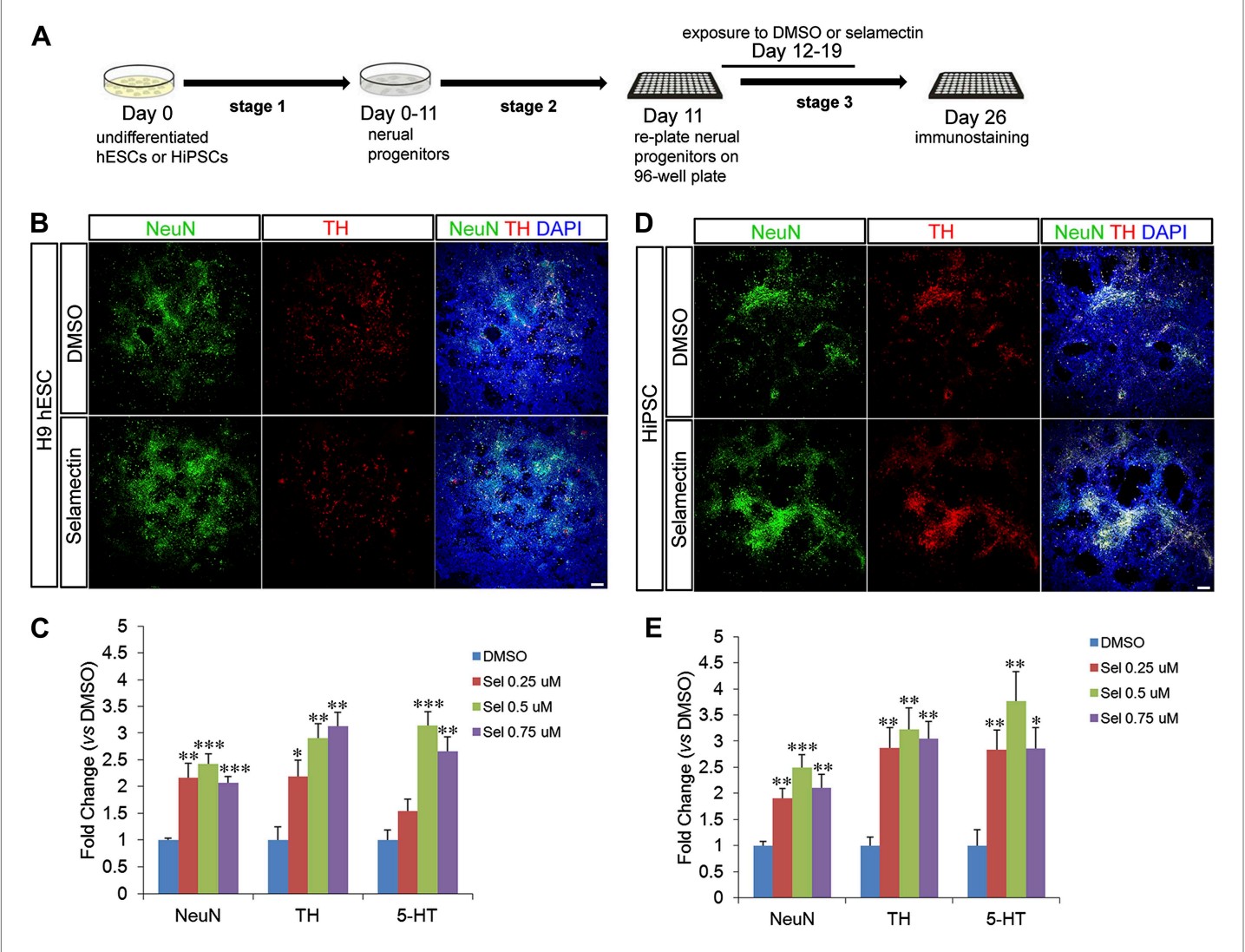

**Figure 10**. Selamectin increases the differentiation of multiple neuronal lineages from human pluripotent stem cells. (**A**) Scheme of the three-stage neuronal differentiation protocol for H9 hESCs and hiPSCs. (**B**) Representative images of NeuN and TH staining in control (DMSO) and selamectin-treated cultures of H9 hESCs. (**C**) Quantification shows increased production of both TH+ and total neurons by selamectin in H9 hESCs. (n = 4, *p<0.05, **p<0.01, ***p<0.001 vs DMSO). (**D**) Representative images of NeuN and TH staining in control (DMSO) and selamectin-treated cultures of hiPSCs. (**E**) Quantification shows increased production of both TH+ and total neurons by selamectin in hiPSCs. (n = 4, *p<0.05, **p<0.01, ***p<0.001 vs DMSO). Scale bar, 100 μm.

(*Song et al., 2012*). Our single neural rosette cell recordings, which reveal the heterogeneous responses among progenitors, provide an explanation as to why the influence of GABA signaling on neurogenesis appears cell type- and context-dependent.

It is worth pointing out that our screen of 2000 compounds was not successful in identifying chemicals that specifically increase DA neuronal production, suggesting that these compounds may be rare, and large compound libraries need to be screened in order to find them. The high throughput capability of our assay will enable such screen to be carried out, and is thus an important future direction.

## Materials and methods

### Cell culture and high throughput chemical screen

The mouse ESC lines E14Tg2a and 46C were used. 46C was a generous gift from Dr Austin Smith, in which GFP was knocked into the *sox1* locus (*Ying and Smith, 2003*). mESCs were cultured in GMEM media (G5154, Sigma, St. Louis, MO) supplemented with glutamine, sodium pyruvate, 0.1 mM MEM

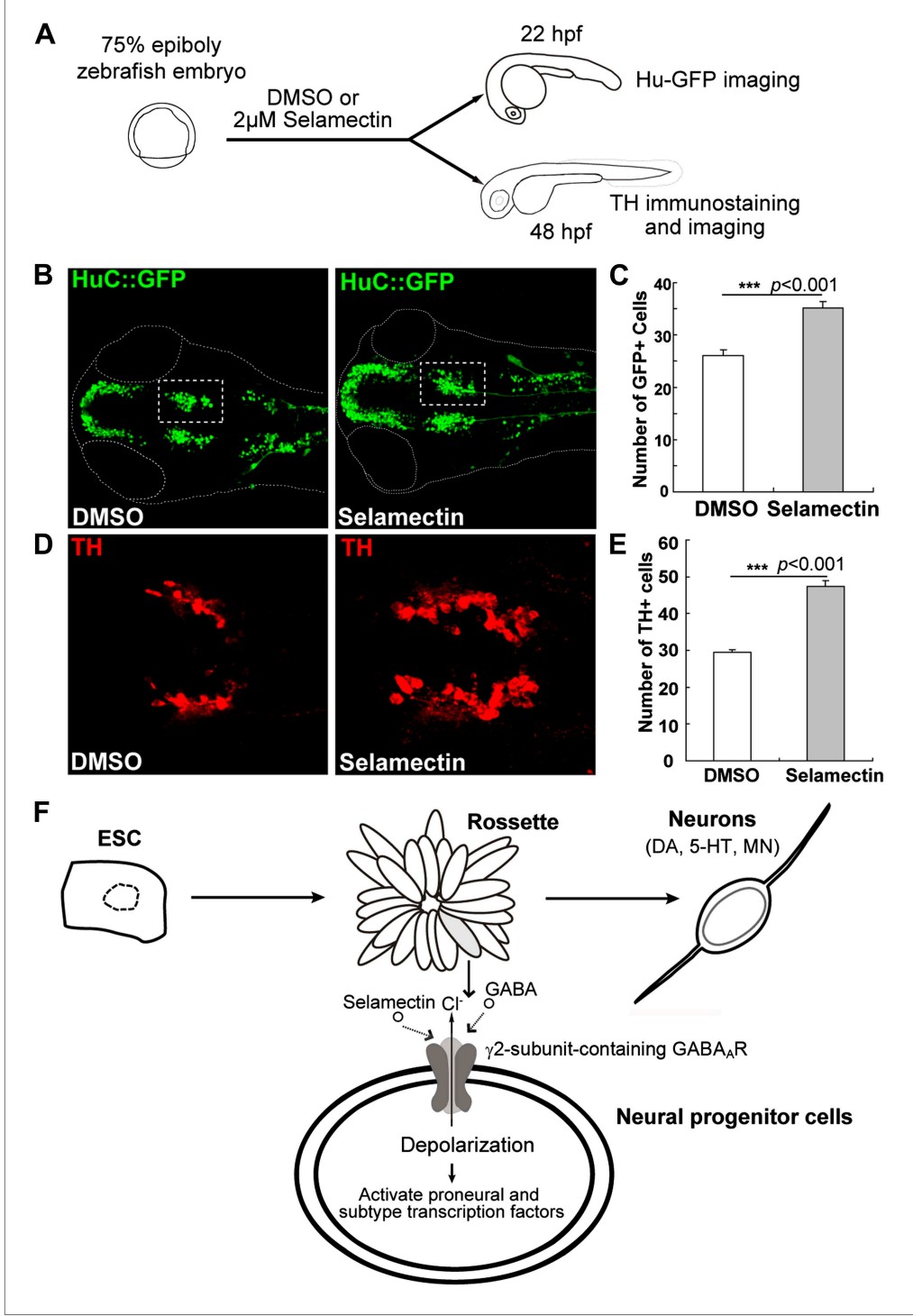

**Figure 11**. Selamectin promotes neurogenesis in vivo in the developing zebrafish brain. (**A**) Scheme of the selamectin treatment on HuC:GFP transgenic zebrafish embryos. (**B**) Representative images show that selamectin (2 µM, 14 hr from 8 hpf to 22 hpf) increases Huc-GFP signal. (**C**) Quantification of the midbrain cluster (boxed) shows a significant difference between two groups (*t*-test, n = 20, p<0.001). (**D**) Representative images show increased DA neurons in selamectin-treated embryos (2 µM for 40 hr from 8 hpf to 48 hpf). (**E**) Quantification of the ventral forebrain DA neurons shows a significant difference between two groups (*t*-test, n = 10, p<0.001). (**F**) A schematic model shows the effect of selamectin on neuronal differentiation from mESCs.

non-essential amino acids, 10% (vol/vol) fetal bovine serum (characterized, Hyclone, Thermo Scientific, Waltham, MA), beta-mercaptoethanol, and 500–1000 units per ml of leukocyte inhibitory factor (ESG1107, Chemicon, Billerica, MA), on gelatinized cell culture surface without feeder cells.

To induce neuronal differentiation, the monolayer differentiation protocol developed by Ying et al. (*Ying et al., 2003*) was used. Briefly, E14 cells was dissociated with Trypsin-EDTA (TE) into single cells and plated onto gelatinized cell culture dish at a density of $1.0 \times 10^4$ cell/cm$^2$ in N2B27 media. Cells were cultured in N2B27 media for 7 days, with media change every other day. On day 7, cells were dissociated with TE again and re-plated onto poly-L-ornithine-laminin coated 96-well plate in N2B27 media, at a density of $2\sim5 \times 10^4$/cm$^2$. Media was changed every 2 days after re-plating.

For high throughput screening, on Day 7, cells were dissociated in TE and re-suspended in fresh N2B27 media. The cell suspension was then dispensed into 96-well micro-clear imaging plates (Greiner cat.no 655956) with the WellMate liquid dispenser (Thermo Matrix), at a density of $1.5 \times 10^4$/ well. These assay plates were incubated in 37°C overnight for cell settling and adherence to the surface. On Day 8, screening compounds were dispensed into assay plates with Biomek FXP Laboratory Automation Workstation (Beckman Coulter, Brea, CA), at a final concentration of 1 µM. Chemical treatment lasted for 3 days from Day 8 to Day 11. On Day 11, chemical treatment was withdrawn via change of media. Cells were cultured in assay plates for additional 3 days until Day 14 before automated immunostaining using PlateMate Plus (Thermo Matrix) and image acquisition with INCell 1000 or 2000 (G.E. Healthcare, Little Chalfont, UK).

The chemical library was obtained from the UCSF SMDC, which is composed of FDA-approved drugs, bioactive compounds, and natural products (Microsource Spectrum Collection).

## Immunocytochemistry

Primary antibodies for immunocytochemistry include: Rabbit-anti-TH (AB152, Millipore, Billerica, MA); Mouse-anti-NeuN, (MAB377, Millipore); mouse anti-sox2 (MAB2018, R&D); Rabbit-anti-Lmx1 (a generous gift from Dr. German, UCSF); mouse-anti-Nestin (MAB353, Chemicon); mouse-anti-islet (DSHB 39.4D5); Rabbit anti-GABA (A2052, Sigma); Rabbit-anti-Olig2 (AB9610, Millipore). After immunostaining, images were taken using the automatic system INCell 1000 or 2000 (GE). 20 field of views on three different channels (For TH, NeuN and DAPI) were taken for each well. Images were analyzed using the INCell Developer software (G.E. Healthcare). The percentage of TH in each well was expressed as a ratio. Fold change of chemical-treated well was calculated relative to the average of DMSO control wells. The percentage of other neuronal types was calculated similarly.

## Pharmacology

The following pharmacological compounds were used in this study: selamectin (01503720, Microsource), avermectin (31732-100 MG, Sigma), taurine (T8691-25 G, Sigma), muscimol (M1523-5MG, Sigma), Chlordiazepoxide (C2517, Sigma), picrotoxin (R284556-50 MG, Sigma), Pentylenetetrazole (P6500-25 G, Sigma), bicuculine (14340-25 MG, Sigma), STY (Strychnine, S0532-5G, sigma). Drugs were prepared as 10 mM stock and diluted to appropriate concentrations as indicated in the text.

## EsiRNA knockdown of GABA$_A$ receptor subunits

The algorithm Deqor is used to design esiRNA, which can be found at http://www.mpi-cbg.de/esiRNA/.

Two rounds of PCR were done to obtain the template for in vitro synthesis of double-stranded RNAs. cDNAs from day 3 mESC-derived neural progenitors was used as template for the first round PCR. Primers for first round PCR begin with T7 'anchor' sequence: 5' GGGCGGGT 3', to which the T7 Anchor primer will anneal in the second round PCR. T7 promoter was incorporated in the primers for the second round PCR. The product from the second round PCR was used as template for in vitro transcription with T7 RNA polymerase. Annealing is done in the same program immediately after in vitro transcription: 1) 37°C, 5.5 hr; 2) 90°C, 3 min; 3) Ramp (0.1°C/s) to 70°C; 4) 70°C, 3 min; 5) Ramp (0.1°C/s) to 50°C; 6) 50°C, 3 min; 7) Ramp (0.1°C/s) to 25°C; 8) 25°C, 3 min. Double-stranded RNA was digested with Shortcut RNAse III (NEB) and purified for transfection. The EsiRNAs targeting different GABA$_A$ receptor subunits were transfected into mESC-derived neural progenitors on day 6 and day 9 using Lipofectamine 2000 Reagent (Invitrogen) following the manufacturer's protocol.

## BrdU incorporation and TUNEL staining

Cells were labeled with 10 µM BrdU for 6 hr before immunostaining. Rat-anti-BrdU antibody (ab6326, Abcam, Cambridge, MA) was diluted 1:2000. An in situ cell death detection kit from Roche (Cat. # 12,156 792 910) was used. Staining was performed following manufacturer's instruction.

## Real time RT-PCR

Total RNA was isolated using TRIzol reagent (Invitrogen) and qPCR was carried out following manufacturer's instructions (Applied Bio-systems). Primer sequences are: *ascl1* (GenBank accession number NM_008553.4), forward, 5'-GAAGCAGGATGGCAGCAGAT-3', reverse,5'-TCGGGCTTAGGTTCAGA CAC-3'; *neuroD1* (GenBank accession number NM_010894.2), forward, 5'-AGCCACGGATCAA TCTTCTC-3', reverse, 5'-ACTGTACGCACAGTGGATTC-3'; *lmx1a* (GenBank accession number NM_033652), forward, 5'- ACCCCTATGGTGCTGAACCT- 3', reverse, 5'- CAGCAACCCTTCACACAGTA -3'; *lmx1b* (GenBank accession number NM_010725), forward, 5'-CTGGGCCAAGAGGTTCTGTC-3', reverse, 5'-GAAGAGCCGAGGAAGCAGTC-3'; *nurr1* (GenBank accession number NM_013613), forward, 5'-CTGGCTATGGTCACAGAGAGACAC-3', reverse, 5'-GGTACCAAGTCTTCCAATTTCAGG-3'; β-*actin* (GenBank accession number NM_007393), forward, 5'-TCCTTCTTGGGTATGGAATCCTG-3' reverse, 5'-GGAGGAGCAATGATCTTGATCTTC-3'. $C_t$ values were the means of triplicate replicates. Each sample was normalized with loading reference β-actin ($\Delta C_t$), and then normalized with expression on Day 8. For relative expression level comparison, the difference in cycle threshold ($\Delta\Delta C_t$) between D11 and D8 was evaluated.

## Clonal analysis of mESC-derived neural progenitor cells

Day 5 mESC culture was transfected with low concentration *pCAG-GFP* plasmid to sparsely label mESC-derived neural progenitors. Transfection was done with Lipofectamine 2000 Reagent (Invitrogen) following the manufacturer's protocol. Briefly, 0.8-μg pCAG-GFP plasmid DNA was used for a single confluent well of a 6-well plate. On day 7, cells were dissociated with Trypsin-EDTA and re-plated onto poly-L-ornithine-laminin coated 96-well plate in N2B27 media, at a density of 2~5 × $10^4$/cm$^2$. Time-lapse live imaging was performed with a 2-hr interval from Day 8 to Day 14, using a third generation automated robotic microscopy system that incorporated several advances over earlier systems (*Arrasate et al., 2004*; *Arrasate and Finkbeiner, 2005*; *Sharma et al., 2012*). Multiple images were taken for each condition and stitched together using a custom designed plugin for the open source image processing package Fiji (*Schindelin et al., 2012*). The cells were fixed for immunostaining on day 14.

## Electrophysiology in mESC-derived neural rosette progenitors

Neural rosettes are recognized because of the cells' characteristic bipolar morphology and their radial floral-like arrangement. We used cell line that expresses GFP under SOX1 promoter. For majority of neural rosettes, we also verified their GFP signals. There are also non-rosette GFP positive cells, among which are new born neurons. New born neurons were identified based on their characteristic morphology, round-shaped cell body. The whole-cell recordings were performed at room temperature. Pipette electrodes (Sutter, Novato, CA) were fabricated using a Sutter P-97 horizontal puller and fire-polished and had final tip resistances of 2–4 MΩ. All recording have been performed using gap free protocol while the membrane potential was holding at −70 mV. The bath solution contained (in mM) NaCl 110, KCl 30, CaCl$_2$ 1.8, MgCl$_2$ 0.5, HEPES 5, and glucose 10, pH adjusted to 7.4 with NaOH. The internal solution for patch recordings contained (in mM) NaCl 10, KCl 130, MgCl$_2$ 0.5, HEPES 5, EGTA 1,and MgATP 5, pH adjusted to 7.3 with KOH. The applications of the activation and inhibition reagents were performed by a pressurized micro-perfusion system. The pressure was typically 7–10 kPa. The stock solutions were made by dissolving the reagents in the bath solution (GABA) or DMSO (bicuculline and selamectin). The stock solutions were kept at −80°C and were diluted to the working concentration using bath solution before each experiment. Unless otherwise indicated, we used 100 μM GABA, 100 μM bicuculline and 8 μM selamectin. The working solutions of bicuculline and selamectin contained up to 0.5% DMSO, therefore the bath solutions containing the corresponding concentration of DMSO were routinely used as control solution prior to the applications of bicuculline or selamectin.

## Differentiation of multiple neuronal lineages from human pluripotent stem cells

The human ESC lines H9 (*Thomson et al., 1998*) and a human iPSC line (a gift from Dr Bruce Conklin) (*Kreitzer et al., 2013*) were used. Stem Cells were cultured on growth factor reduced Matrigel (BD Biosciences, Franklin Lakes, NJ) in mTeSR1 media (Stemcell Technologies, Vancouver, Canada) with the media changed daily. To initiate differentiation, H9 and WTC-10 cells were plated at a density of 2 × $10^4$ cells/cm$^2$ in N2B27 media (DMEM/F12:Neurobasal [1:1], N2 supplement (1:100), B27

supplement without vitamin A (1:50), Glutamax, Insulin (20 µg/ml), beta-mercaptoethanol (110 µM), BSA Fraction V (20 µg/ml), bFGF (20 ng/ml) supplemented with Rock Inhibitor Y-27632 (10 µM, Millipore)). Media was changed with fresh N2B27 media every other day until Day 11. Cells were plated at $3 \times 10^4$ cells/cm$^2$ in N2B27 media supplemented with Rock inhibitor on Day 11. Media was changed on Day 12 to neuronal differentiation media (Neurobasal, B27 without vitamin A (1:50), BDNF (20 ng/ml, Peprotech, Rocky Hill, NJ), GDNF (10 ng/ml, Peprotech), cAMP (500 µM, Sigma Aldrich), Ascorbic Acid (200 µM, Sigma Aldrich)). Cells were treated with DMSO or Selamectin from Day 12 to Day 19, and were fixed on Day 26 for immunostaining. All reagents were purchased from LIfe Technologies, unless otherwise stated.

### In vivo zebrafish treatment and immunostaining

A transgenic line (Hu-GFP) marking nascent neurons was used. At 10 hpf, embryos were de-chorionated with forceps and transferred to a glass vial with 3 ml Embryo Solution, and Selamectin was added into solution to a final concentration of 2 µM. Embryos were incubated at 28°C until 22 hpf and then fixed with 4% PFA and mounted onto slides for confocal imaging.

Embryos were also incubated at 28°C until 48 hpf to evaluate the effect of selamectin on TH differentiation (selamectin treatment lasted from 8 hpf to 48 hpf). At 48 hpf, embryos were stained with anti-TH antibody (custom made, 1:1000).

## Acknowledgements

We thank Dr A Smith for the 46C ESC line, Dr Bruce Conklin for the hiPSC line, Dr K Ye for DHF and DOG compounds, Drs Michael Keiser and Brian Shoichet for discussions, Dr B Lu and Guo lab members for helpful comments.

## Additional information

### Funding

| Funder | Grant reference number | Author |
| --- | --- | --- |
| National Institutes of Health | DA023904 | Yaping Sun, Zhiqiang Dong, Jisong Peng, Su Guo |
| CIRM | GUO-RS1-00215-1 | Yaping Sun, Zhiqiang Dong, Jisong Peng, Su Guo |
| Howard Hughes Medical Institute | | Taihao Jin, Lily Y Jan |
| National Institutes of Health | MH065334 | Taihao Jin, Lily Y Jan |
| UCSF QB3-Malaysia Program | | Kean-Hooi Ang, Michelle R Arkin |
| National Institutes of Health | R01CA102321, P01CA081403 | Miller Huang, Michelle R Arkin |
| American Cancer Society | PF-13-295-01-TBG | Miller Huang |
| National Institutes of Health | U24 NS078370 | Kelly M Haston, Steven Finkbeiner |
| CIRM | RB4-06079 | Kelly M Haston, Steven Finkbeiner |
| Chinese Ministry of Science and Technology | 2013CB945301 | Tao P Zhong |

The funders had no role in study design, data collection and interpretation, or the decision to submit the work for publication.

### Author contributions

YS, ZD, Conception and design, Acquisition of data, Analysis and interpretation of data, Drafting or revising the article; TJ, MH, Acquisition of data, Analysis and interpretation of data, Drafting or revising the article; K-HA, TPZ, SF, WAW, MRA, Drafting or revising the article, Contributed unpublished essential data or reagents; KMH, JP, Acquisition of data, Drafting or revising the article; LYJ, Analysis

and interpretation of data, Drafting or revising the article; SG, Conception and design, Analysis and interpretation of data, Drafting or revising the article

### Ethics

Animal experimentation: This study was performed in strict accordance with the recommendations in the Guide for the Care and Use of Laboratory Animals of the National Institutes of Health. All of the animals were handled according to approved institutional animal care and use committee (IACUC) protocols (#AN083282) of the University of California, San Francisco.

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
