## [Decision Letter]

Thank you for choosing to send your work entitled “Imaging-based chemical screening reveals activity-dependent neural differentiation in mouse embryonic stem cells” for consideration at *eLife*. Your article has been favorably evaluated by a Senior editor and 3 reviewers, one of whom is a member of our Board of Reviewing Editors.

The Reviewing editor has assembled the following comments to help you prepare a revised submission.

1) As already discussed by the authors themselves, the effects of selamectin should be tested for the neuronal differentiation of human pluripotent stem cells.

2) It is likely that selamectin induced mESC differentiation into multiple neural lineages. However, it is not clear whether this effect is due to the heterogeneity of neural progenitor cells derived from mESCs. In the former case, selamectin could have just induced the terminal differentiation of precursor cells already committed toward particular neuronal lineages. Alternatively, selamectin must have induced the mESC differentiation into multiple neural lineages by different mechanisms. This issue should be investigated by clonal analysis of mESC-derived neuronal progenitor cells to examine the effects of selamectin.

3) One of the primary conclusions of the paper is that selamectin is functioning as a GABAA agonist. The most compelling evidence for this was pharmacological: the structurally unrelated GABA agonist muscimol had a similar effect, while GABA antagonists reduced the effect of selamectin. It would likely strengthen the case if there were some non-pharmacological (e.g., genetic) evidence.

4) In the present assay, authors used TH as a readout marker to address neuronal subtype (dopaminergic neuron)-directed differentiation, but their hit compounds increased the differentiation of multiple neuronal lineages without selectivity for specific neuronal subsets. In this point, their screening was not successful in getting subtype-selective compounds. Authors should make comments for this criticism and should discuss in the text.

5) Because the present assay focuses on ratio of TH positivity instead of cell number, compounds that reduce viability of a non-neuronal cell type could also appear to be increasing neural differentiation.

---

## [Author Response]

*1) As already discussed by the authors themselves, the effects of selamectin should be tested for the neuronal differentiation of human pluripotent stem cells*.

We have performed the experiment using both the human embryonic stem cells (H9) and the human induced pluripotent stem cells (hiPSC). The results as shown in Figure 10 indicate that selamectin promotes neuronal differentiation from human pluripotent stem cells.

*2) It is likely that selamectin induced mESC differentiation into multiple neural lineages. However, it is not clear whether this effect is due to the heterogeneity of neural progenitor cells derived from mESCs. In the former case, selamectin could have just induced the terminal differentiation of precursor cells already committed toward particular neuronal lineages. Alternatively, selamectin must have induced the mESC differentiation into multiple neural lineages by different mechanisms. This issue should be investigated by clonal analysis of mESC-derived neuronal progenitor cells to examine the effects of selamectin*.

We have carried out clonal analysis in mESC cultures by performing time-lapse live imaging of neuronal progenitor cells sparsely labeled with GFP, followed by immunostaining of progenitor and neuronal markers. The results as shown in Figure 9 indicate that selamectin did not change cell cycle length or the clone size. Rather, it increased differentiated neurons at the expense of proliferative progenitors. This, together with our previous observation that selamectin decreases proliferation and increases the expression of proneural and lineage-associated transcription factors, suggests that it promotes progenitor cell cycle exit toward terminal differentiation.

*3) One of the primary conclusions of the paper is that selamectin is functioning as a GABAA agonist. The most compelling evidence for this was pharmacological: the structurally unrelated GABA agonist muscimol had a similar effect, while GABA antagonists reduced the effect of selamectin. It would likely strengthen the case if there were some non-pharmacological (e.g., genetic) evidence*.

We have employed EsiRNAs to genetically impair the activity of various GABA-A receptor subunits. The results as shown in Figure 6 indicate that impairment of the γ2 subunit (encoded by the *gabarg2* gene) abolishes the effect of selamectin, thus providing genetic evidence that selamectin exerts its effect through the GABA-A receptor.

*4) In the present assay, authors used TH as a readout marker to address neuronal subtype (dopaminergic neuron)-directed differentiation, but their hit compounds increased the differentiation of multiple neuronal lineages without selectivity for specific neuronal subset. In this point, their screening was not successful in getting subtype-selective compounds. Authors should make comments for this criticism and should discuss in the text*.

We have discussed this point (see last paragraph of the Discussion).

*5) Because the present assay focuses on ratio of TH positivity instead of cell number, compounds that reduce viability of a non-neuronal cell type could also appear to be increasing neural differentiation*.

We agree that it is a formal possibility. However, reduced viability of non-neuronal cell types would be detectable as it would lead to a significant reduction of total cell numbers in the well compared to controls. Furthermore, clonal analysis, as we have shown in the revised manuscript, will be able to further discern this possibility.